# Variational Inference for Graph Convolutional Networks in the Absence of Graph Data and Adversarial Settings

**Pantelis Elinas**[*]
pantelis.elinas@data61.csiro.au
CSIRO's Data61

**Edwin V. Bonilla**[*]
edwin.bonilla@data61.csiro.au
CSIRO's Data61 & The University of Sydney

**Louis C. Tiao**
louis.tiao@sydney.edu.au
The University of Sydney & CSIRO's Data61

## Abstract

We propose a framework that lifts the capabilities of graph convolutional networks (GCNs) to scenarios where no input graph is given and increases their robustness to adversarial attacks. We formulate a joint probabilistic model that considers a prior distribution over graphs along with a GCN-based likelihood and develop a stochastic variational inference algorithm to estimate the graph posterior and the GCN parameters jointly. To address the problem of propagating gradients through latent variables drawn from discrete distributions, we use their continuous relaxations known as Concrete distributions. We show that, on real datasets, our approach can outperform state-of-the-art Bayesian and non-Bayesian graph neural network algorithms on the task of semi-supervised classification in the absence of graph data and when the network structure is subjected to adversarial perturbations.

## 1 Introduction

Graphs represent the elements of a system and their relationships as a set of nodes and edges, respectively. By exploiting the inter-dependencies of these elements, many applications of machine learning have achieved significant success, for example in the areas of social networks [15], document classification [25, 40] and bioinformatics [12]. In particular, motivated by the incredible success of convolutional neural networks [CNNs, 26] on regular-grid data, researchers have generalized some of their fundamental properties (such as their ability to learn local stationary structures efficiently) to graph-structured data [7, 17, 10]. These approaches mainly focused on exploiting feature dependencies explicitly defined by a graph in an analogous way to how convolutional neural networks (CNNs) model long-range correlations through local interactions across pixels in an image. The seminal work by [25] leveraged these ideas to model *dependencies across instances* (instead of features) to be able to incorporate knowledge of the instances' relationships in a semi-supervised learning setting, going beyond the standard i.i.d. assumption.

In this work we focus precisely on the problem of semi-supervised classification based on the method developed in [25], which is now commonly referred to as graph convolutional networks (GCNs). These networks can be seen as a first-order approximation of the spectral graph convolutional networks developed by [10], which itself built upon the pioneering work of [7, 17]. The great popularity of graph convolutional networks (GCNs) is mainly due to their practical performance

---

[*]Joint first author.

as, at the time it was published, it outperformed related methods by a significant margin. Another practical advantage of using GCNs is their relatively simple propagation rule, which does not require expensive operations such as eigen-decomposition.

However, one of the main assumptions underlying GCNs is that the given graph is helpful for the task at hand and that the corresponding edges are highly reliable. This is generally not true in practical applications, as the given graph may be (i) noisy, (ii) loosely related to the classification problem of interest or (iii) built in an ad hoc basis using, e.g., side information. Although these settings have been addressed by previous work independently [see e.g. 13, 48], in practice, it is difficult to incorporate this type of uncertainty over the graph using the original GCN framework in a principled way and the performance of the method degrades significantly with increasingly noisy graphs.

Consequently, in this paper we propose a framework that lifts GCN's capabilities to handle the more challenging cases of learning in the absence of an input graph and dealing with highly-effective adversarial perturbations such as those proposed recently by [4]. We achieve this by formulating a joint probabilistic model that places a prior over the graph structure, as given by the adjacency matrix, along with a GCN-based conditional likelihood. This, however, poses significant inference challenges as estimating the posterior over the adjacency matrix under a highly non-linear likelihood (as given by the GCN's output) is intractable.

Thus, to estimate the graph posterior we resort to approximations via stochastic variational inference (VI). Nevertheless, even in the approximate inference world, carrying out posterior estimation over a very large discrete combinatorial space can prove extremely hard. We adopt a simple but effective relaxation over both the prior and the approximate posterior using Concrete distributions [28], which allows us to propagate gradients in the corresponding stochastic computational graph. Our experiments show that we can outperform state-of-the-art Bayesian and non-Bayesian graph neural network algorithms in the task of semi-supervised classification (i) in the absence of graph data; (ii) when the network structure is subjected to adversarial perturbations and (iii) when considering the ground truth graphs. Our results and analyses indicate that our framework does indeed learn new graph representations by turning on/off connections so as to improve performance on the given task.

## 1.1 Related work

Most graph neural network frameworks can be seen from a more general perspective under the unifying mixture models network [MONET, 29]. For details of graph neural network approaches the reader is referred to the excellent related work in [40] and, more generally, to the surveys in [47, 43]. With regards to uncertain graphs, [5] propose a graph-anonymization technique that injects uncertainty in the existence of edges of the graph. [9] propose a method that models the probability of a node belonging to a particular class as a function of the uncertainty in the edges related to that node. However, such an approach does not build upon state-of-the-art graph networks nor develop a fully coherent probabilistic model over the parameters of the network. Following a different methodology, [19] deal with the problem of uncertain graphs via embeddings by constructing a proximity matrix given the uncertain graph and applying matrix factorization to get the embedded representation. The embeddings are then used in supervised learning tasks. Unlike our method, this is a two-step procedure where the prediction task is separate from uncertainty modeling and representation learning.

In terms of probabilistic approaches, [46] propose a model for GCNs where the graph is considered as a realization of mixed membership stochastic block models [2]. However, despite their method being referred to as Bayesian, it parameterizes the *true posterior* over the graph directly and this posterior is not dependent on the observed data (neither features or labels). Thus, it can be seen more like an ensemble GCN. Targeting the low-labeled-data regime, [31] propose a method for semi-supervised classification with Gaussian process (GP) priors, where the parameters of a robust-max likelihood for a node are given by the average of the GP values over its 1-hop neighborhood. Their results indicate that their method can outperform GCNs in active learning settings.

Concerning graph structure learning, similarly to our work, [23] use variational autoencoders during inference to learn this structure. However, their focus is very different to ours as they address the problem of learning the interactions between components in a dynamical system given their trajectories (i.e., the entities evolve over time) in an unsupervised way. More recently, [13] develop a method for learning the graph structure using a generative model and estimate its parameters via

bilevel optimization. However, unlike our approach, their method is not Bayesian and it does not allow for either the incorporation of prior knowledge or estimation of the full posterior over the graph. To elaborate on this point, its initial edge probabilities use a deterministic distribution (see their Algorithm 1) and do not play an explicit role in the objective function being optimized, i.e. there is no prior constraining the search-space over graphs. Using different initial probabilities will not achieve a similar effect to that obtained in our joint probabilistic framework. Finally, [24] propose the variational graph autoencoder, a probabilistic framework that learns latent representations for graphs but, unlike our work, is designed for the task of link prediction.

## 2 Bayesian graph convolution models

Let $\mathbf{X} \in \mathbb{R}^{N \times D}$ be a set of $D$-dimensional features representing $N$ instances and $\mathbf{Y} = \{\mathbf{y}_n\}$ be their corresponding labels, some of which are observed and others unobserved and $\mathbf{y}_n \in \{0,1\}^C$ is one-hot-encoded. The goal of semi-supervised classification is to leverage the labeled and unlabeled data in order to predict the unobserved labels. In this paper we are interested in doing so by explicitly exploiting the dependencies among datapoints as given by an undirected graph $\mathcal{G} = (\mathcal{V}, \mathcal{E})$ with $N$ nodes $v_i \in \mathcal{V}$, edges $(v_i, v_j) \in \mathcal{E}$ and binary adjacency matrix $\mathbf{A} \in \{0,1\}^{N \times N}$.

**GCN's basic propagation rule.** To do this, we consider the popular GCN models [25], which can be seen as first-order approximations to more general (but computationally costly) spectral graph convolutional networks [10]. For a signal $\mathbf{X}$, [25] showed that one can write a convolved signal matrix as $\tilde{\mathbf{D}}^{-\frac{1}{2}} \tilde{\mathbf{A}} \tilde{\mathbf{D}}^{-\frac{1}{2}} \mathbf{X} \mathbf{W}$, where $\mathbf{W}$ is a matrix of filter parameters, $\tilde{\mathbf{A}} = \mathbf{A} + \mathbf{I}_N$ is the adjacency matrix of the graph augmented with self-loops and $\tilde{\mathbf{D}}$ is the corresponding degree matrix with $\tilde{\mathbf{D}}_{ii} = \sum_{j=1}^{N} \tilde{\mathbf{A}}_{ij}$. This convolved signal matrix constitutes the basic operation in GCNs.

**Composition of graph convolutions.** Thus, we can define compositions of these approximate spectral graph convolutions by the recurrence relation, $\mathbf{f}^{(0)}(\mathbf{X}, \mathbf{A}) = \mathbf{X}$, $\mathbf{f}^{(l+1)}(\mathbf{X}, \mathbf{A}) = h^{(l+1)}\left(\hat{\mathbf{A}} \mathbf{f}^{(l)}(\mathbf{X}, \mathbf{A}) \mathbf{W}_l\right)$, where $\hat{\mathbf{A}} \equiv \tilde{\mathbf{D}}^{-\frac{1}{2}} \tilde{\mathbf{A}} \tilde{\mathbf{D}}^{-\frac{1}{2}}$, $\tilde{\mathbf{A}}$ and $\tilde{\mathbf{D}}$ defined as above and $h^{(l)}(\cdot)$ is a nonlinear activation function of the $l$-th layer, typically the element-wise rectified linear unit (RELU), $\texttt{relu}(\cdot) = \max(0, \cdot)$. GCN uses these types of compositions to define a neural network architecture for semi-supervised classification, where the activation for the final layer $h^{(L)}(\cdot)$ is the row-wise softmax function. Each layer is parameterized by $\mathbf{W}_l$, a $Q^{(l)} \times Q^{(l+1)}$ matrix of weights, where $Q^{(l)}$ is the number of hidden units for layer $l$, with $Q^{(0)} = D$ and $Q^{(L)} = C$. We will denote the GCN parameters with $\boldsymbol{\theta} = \{\mathbf{W}_l\}_{l=1}^{L}$.

**Two-layer GCN.** In this paper we will focus on two-layer architectures, hence the output of the GCN at the last layer, i.e. $\mathbf{\Pi} \equiv \mathbf{f}^{(L)}(\mathbf{X}, \mathbf{A})$ is given by:

$$\mathbf{\Pi} = \texttt{softmax}\left(\hat{\mathbf{A}} \texttt{relu}\left(\hat{\mathbf{A}} \mathbf{X} \mathbf{W}_0\right) \mathbf{W}_1\right), \tag{1}$$

where $\mathbf{\Pi}$ is an $N \times C$ matrix of probabilities for all nodes and classes.

When the given graph is highly reliable, GCNs trained with the proposed cross-entropy minimization method can yield state-of-the-art classification results [25]. However, one would like to lift GCN capabilities to scenarios where there is *no input graph* or make them robust to cases when the graph has been perturbed *adversarially*. In order to address these problems, we propose a joint probabilistic model that considers the graph parameters as random variables and develop a stochastic variational inference algorithm to estimate the posterior over these parameters. This posterior is then used in conjunction with the GCN parameters for making predictions over the unlabeled instances.

### 2.1 Likelihood

Our likelihood model assumes that, conditioned on all the features and the graph adjacency matrix, the observed labels $\mathbf{Y}^o$ are conditionally independent, i.e.,

$$p_{\boldsymbol{\theta}}(\mathbf{Y}^o \,|\, \mathbf{X}, \mathbf{A}) = \prod_{\mathbf{y}_n \in \mathbf{Y}^o} p_{\boldsymbol{\theta}}(\mathbf{y}_n \,|\, \mathbf{X}, \mathbf{A}) \quad \text{with} \quad p_{\boldsymbol{\theta}}(\mathbf{y}_n \,|\, \mathbf{X}, \mathbf{A}) = \text{Cat}(\mathbf{y}_n \,|\, \boldsymbol{\pi}_n), \tag{2}$$

where $\text{Cat}(\mathbf{y}_n | \boldsymbol{\pi}_n)$ denotes a Categorical distribution over $\mathbf{y}_n$ with parameters $\boldsymbol{\pi}_n$ being the $n$-th row of the $N \times C$ probability matrix $\mathbf{\Pi}$ obtained from a GCN with $L$ layers, i.e., $\mathbf{\Pi} = \mathbf{f}^{(L)}(\mathbf{X}, \mathbf{A})$. As

before, $\boldsymbol{\theta}$ denotes the GCN's weight parameters. One of the fundamental differences of our approach with standard GCNs is that we consider a prior over graphs that is constructed using the observed (but potentially noisy or unreliable) adjacency matrix.

## 2.2 Prior over graphs

We consider random graph priors of the form

$$p(\mathbf{A}) = \prod_{ij} p(A_{ij}), \text{ with } p(A_{ij}) = \text{Bern}(A_{ij} \,|\, \rho_{ij}^o), \tag{3}$$

where $\text{Bern}(A_{ij} \,|\, \rho_{ij}^o)$ is a Bernoulli distribution over $A_{ij}$ with parameter $\rho_{ij}^o$. Our prior is constructed given an observed (auxiliary) graph $\bar{\mathbf{A}}$ but, for simplicity in the notation, we omit this conditioning here and in all related distributions. This prior can be constructed in various ways so as to encode our beliefs about the graph structure and about how much this structure should be trusted a priori for our semi-supervised classification task. For example, we have found that the construction $\rho_{ij}^o = \bar{\rho}_1 \bar{\mathbf{A}}_{ij} + \bar{\rho}_0 (1 - \bar{\mathbf{A}}_{ij})$, with $0 < \bar{\rho}_1, \bar{\rho}_0 < 1$ being hyper-parameters, works very well in practice as it gives just enough flexibility to encode our degree of belief on the absence and presence of links separately.

In the adversarial setting, i.e., when the adjacency matrix of the given graph is altered through adversarial perturbations, $\bar{\mathbf{A}}$ is simply given by the perturbed matrix. In the case when no input graph is provided, $\bar{\mathbf{A}}$ can be obtained through a k-nearest neighbor graph (K-NNG). In other words, given a distance function $d(\cdot, \cdot)$, $\bar{\mathbf{A}}_{i,j} = 1$ iff $d(\mathbf{x}_i, \mathbf{x}_j)$ is among the $k$ smallest distances from $\mathbf{x}_i$ to all other instances and $\bar{\mathbf{A}}_{i,j} = 0$ otherwise.

# 3 Graph structure inference

Our goal is to carry out joint inference over the GCN parameters and the graph structure as given by the adjacency matrix. Since the main additional component of our approach is to consider a prior over the adjacency matrix, we focus on the estimation of the posterior over this matrix given the observed data, i.e. $p(\mathbf{A} \,|\, \mathbf{X}, \mathbf{Y}^o) \propto p_{\boldsymbol{\theta}}(\mathbf{Y}^o \,|\, \mathbf{X}, \mathbf{A}) p(\mathbf{A})$ where the likelihood and prior terms are given by eqs. 2 and 3, respectively.

Computation of this posterior is analytically intractable due to the highly non-linear nature of the likelihood so we resort to approximate posterior inference methods. Given the high-dimensional nature of the posterior over $\mathbf{A}$, we focus on compact representations of the posterior via VI [21]. Generally, VI methods entail the definition of an approximate posterior (variational) distribution and the estimation of its parameters via the maximization of the so-called evidence lower bound (ELBO), a procedure that is known to be equivalent to minimizing the Kullback-Leibler (KL) divergence between the approximate and true posterior distributions.

## 3.1 Variational distribution: free parameterization and continuous relaxations

Similarly to the prior definition, our approximate posterior is of the form

$$q_{\boldsymbol{\phi}}(\mathbf{A}) = \prod_{ij} q_{\boldsymbol{\phi}}(A_{ij}), \text{ with } \quad q_{\boldsymbol{\phi}}(A_{ij}) = \text{Bern}(A_{ij} \,|\, \rho_{ij}), \rho_{ij} > 0, \tag{4}$$

where, henceforth, we use $\boldsymbol{\phi}$ to denote all the parameters of the variational posterior. In the case where $\rho_{ij}$ are free parameters then $\boldsymbol{\phi} = \{\rho_{ij}\}$. We refer to this approach as the *free* parameterization. The variational distribution defined in eq. 4 naturally models the discrete nature of the adjacency matrix $\mathbf{A}$. Our goal is then to estimate the parameters $\boldsymbol{\phi}$ of the posterior $q_{\boldsymbol{\phi}}(\mathbf{A})$ via maximization of the evidence lower bound (ELBO). For this purpose we can use the so-called score function method [33], which provides an unbiased estimator of the gradient of an expectation of a function using Monte Carlo (MC) samples. However, it is now widely accepted that, because of its generality, the score function estimator can suffer from high variance [34].

Therefore, as an alternative to the score function estimator, we can use the so-called re-parameterization trick [22, 35], which generally exhibits lower variance. Unfortunately, the re-parameterization trick is not applicable to discrete distributions so we need to resort to continu-

ous relaxations. In this work we use Concrete distributions as proposed by [20, 28]. In particular, we denote our binary Concrete posterior distribution with location parameters $\lambda_{ij} > 0$ and temperature $\tau > 0$ as $q_\phi(A_{ij}) = \text{BinConcrete}(A_{ij} \,|\, \lambda_{ij}, \tau)$. Analogously, as discussed in [28], in order to maintain a lower bound during variational inference we also relax our prior so that $p(A_{ij}) = \text{BinConcrete}(A_{ij} \,|\, \lambda_{ij}^o, \tau_o)$. In this case the variational parameters are the parameters of the Concrete distribution $\phi = \{\lambda_{ij}\}$.

Our experiments in § 4 focus on freely-parameterized variational posteriors using continuous relaxations via Concrete distributions. However, we have also analyzed the performance of discrete approaches along with low-rank parameterizations such as those used by [24]. These analyses, detailed in the supplement, show that our approach is superior and can be explained by recent results regarding the severe limitations of low-rank representations of graphs [37].

## 3.2 Evidence lower bound

It is easy to show that we can write the ELBO as

$$\mathcal{L}_{\text{ELBO}}(\phi) = \bar{\ell}_\phi\left(\mathbf{Y}^o, \mathbf{X}, \mathbf{A}\right) - \text{KL}\left[q_\phi(\mathbf{A}) \,\|\, p(\mathbf{A})\right], \qquad (5)$$

where $\bar{\ell}_\phi\left(\mathbf{Y}^o, \mathbf{X}, \mathbf{A}\right) \equiv \mathbb{E}_{q_\phi(\mathbf{A})} \log p_\theta(\mathbf{Y}^o \,|\, \mathbf{X}, \mathbf{A})$ is the expected log likelihood (ELL), i.e. the expectation of the conditional likelihood over the approximate posterior, and $\text{KL}\left[q_\phi(\mathbf{A}) \,\|\, p(\mathbf{A})\right]$ is the KL divergence between the approximate posterior and the prior. The ELBO when using the Concrete relaxations under a more numerically stable parameterization (see the supplement for details) is given by

$$\mathcal{L}_{\text{ELBO}}(\phi) = \mathbb{E}_{g_{\phi,\tau}(\mathbf{B})}\left[\log p_\theta(\mathbf{Y}^o \,|\, \mathbf{X}, \sigma(\mathbf{B})) - \log \frac{g_{\phi,\tau}(\mathbf{B})}{f_{\tau_o}(\mathbf{B})}\right], \qquad (6)$$

where

$$g_{\phi,\tau}(B_{ij}) = \text{Logistic}\left(B_{ij} \,\Big|\, \frac{\log \lambda_{ij}}{\tau}, \frac{1}{\tau}\right), \quad f_{\tau_o}(B_{ij}) = \text{Logistic}\left(B_{ij} \,\Big|\, \frac{\log \lambda_{ij}^o}{\tau_o}, \frac{1}{\tau_o}\right), \qquad (7)$$

$\sigma(\mathbf{B})$ computes the entrywise logistic sigmoid function over $\mathbf{B}$; $\text{Logistic}(B \,|\, \mu, s)$ denotes a Logistic distribution with location $\mu$ and scale $s$ and the distributions $g_{\phi,\tau}(\mathbf{B})$ and $f_{\tau_o}(\mathbf{B})$ factorize over the entries of $\mathbf{B}$. The expectation $E_{g_{\phi,\tau}(\mathbf{B})}$ is estimated using $S$ samples from the re-parameterized posterior, which can be obtained using eq. 9 below. Estimation of the variational parameters $\phi$ is done via gradient-based optimization of the ELBO with the gradients obtained by automatic differentiation.

## 3.3 Predictions

The posterior predictive distribution of the latent labels, given our factorized assumption of the conditional likelihood in eq. 2, can be obtained as

$$p(\mathbf{Y}^u \,|\, \mathbf{Y}^o, \mathbf{X}) = \sum_{\mathbf{A}} p_\theta(\mathbf{Y}^u \,|\, \mathbf{X}, \mathbf{A}) p(\mathbf{A} \,|\, \mathbf{Y}^o, \mathbf{X}) \approx \frac{1}{S} \sum_{s=1}^{S} p_\theta(\mathbf{Y}^u | \mathbf{X}, \mathbf{A}^{(s)}), \qquad (8)$$

where $\mathbf{A}^{(s)}$ is a sample from the posterior $q_\phi(\mathbf{A})$, $S$ is the total number of samples and $p_\theta(\mathbf{Y}^u \,|\, \mathbf{X}, \mathbf{A})$ is the GCN-likelihood given in eq. 2. These samples can be obtained as:

$$U \sim \text{Uniform}(0, 1), \quad A_{ij}^{(s)} = \sigma\left(\frac{\log \lambda_{ij} + \log U - \log(1 - U)}{\tau}\right), \qquad (9)$$

where, as before, $\{\lambda_{ij}\}$ are the estimated parameters of the posterior and $\tau$ is the posterior temperature.

## 3.4 Computational complexity

We require to compute $\mathcal{O}(N^2)$ individual KL divergences, which can be trivially parallelized. While for a discrete posterior these individual KL terms can be computed exactly (as shown in the supplement), for the continuous relaxation we need to resort to MC estimation over $S$ samples. Aggregation over samples can also be parallelized straightforwardly. Computing the ELL using a 2-layer GCN as in eq. 1 requires $\mathcal{O}(NDQ + S(NQC + N^2Q + N^2C))$ for the continuous case. However, in the

discrete case it only requires doing a forward pass over the standard GCN architecture $S$ times, hence being linear in the number of edges, i.e. $\mathcal{O}(S|\mathcal{E}|DQC)$, where $|\mathcal{E}|$ is the expected number of edges sampled from the posterior, assuming sparse-dense matrix multiplication is exploited. In order to reduce the number of parameters and allow for mini-batch training, our approach can be combined with other methods such as cluster-GCN [8]. We present an example of this as well as more details of our method's computational complexity in the supplement.

## 4 Experiments

In this section we describe the experiments carried out to evaluate our method, which we will refer to as variational graph convolutional network (VGCN)[2]. We will focus on its relaxed version when using the free parameterization with binary Concrete prior and posterior distributions, which we found to be much more stable during training than when using the discrete version and outperformed the low-rank parameterization under latent-dimensionality constraints (see the supplement for details). We will start by analyzing the scenario when no input graph is given and will refer to this as the *no-graph* case. This is motivated by the fact that in many practical applications graphs are created in an ad hoc basis based on side information or node features [17]. Then we present results on the robustness of the algorithm when there is an input graph associated with the given dataset and such a graph is subjected to adversarial perturbations. We refer to this scenario as the *adversarial setting*. We conclude the section with an analysis of the estimated posterior distribution and by showcasing the performance of our method when using the ground truth (unperturbed) graphs.

### 4.1 Experimental set-up

**Datasets.** We use two citation-network datasets (CORA and CITESEER) and an additional graph of political blogs (POLBLOGS). In the citation networks the nodes represent documents and their links refer to citations between documents. We construct an undirected graph based on these citation links so as to obtain a binary adjacency matrix. Each document is characterized by a set of features, which are either bag-of-words (BOW) or term frequency-inverse document frequency (TF-IDF) for CORA and CITESEER, respectively. The class for each document is given by their subject and our goal is to predict this for a subset of unlabeled documents. The POLBLOGS dataset is a network of political blogs first introduced in [1]. For our experiments, we use the dataset as published in [4]. Nodes represent blogs and two nodes are connected if a blog links to another from anywhere on the landing page. Each node is labeled as one of two classes based on the blog's political affiliation (conservative vs liberal). No node features are available and so we use an identity matrix as is common practice. Details of these datasets are given in the supplement.

**Training details.** For the citation networks we used a similar setting to that in [44] where 20 labeled examples per class are used for training, 1,000 examples are used for testing and the rest are used as unlabeled data. We are very much aware of the potential difficulties when using fixed training datasets for evaluation in machine learning, in general, and in particular in graph neural networks [38]. However, we believe our experiments introduce enough additional randomization so that the results can be considered as reliable. Firstly, we adopt the set-up in the recently proposed work of [13], where the original splits are augmented so as to include 50% of the validation set. Furthermore, in the no-graph case, we generate 10 different versions of each dataset and also construct K-NNGs using $k = \{10, 20\}$ and the cosine and Minkowski distances. In the adversarial setting, also using the augmented datasets, we explore 7 different noise-level aggregations and replicate each experiment 10 times. To alleviate the phenomenon of the variational posterior collapsing to the prior and thereby failing to learn latent representations that explain the data [6, 11], we dampen the effect of the KL regularization term in the ELBO by scaling via a parameter $\beta < 1$ [18, 3]. This and other hyper-parameters were tuned using cross-validation, see the supplement for details.

**Baselines and performance metrics.** We compare against standard GCN [25], sample-and-aggregate[3] [GRAPHSAGE, 15] and graph attention networks [GATs, 40], which are all competitive graph neural network algorithms that assume a noise-free input graph is given. Additionally, we benchmark our algorithm against the learning discrete structures (LDS) framework of [13] which, like ours, attempts to learn a graph generative model for GCNs (see § 1.1 for more details). We

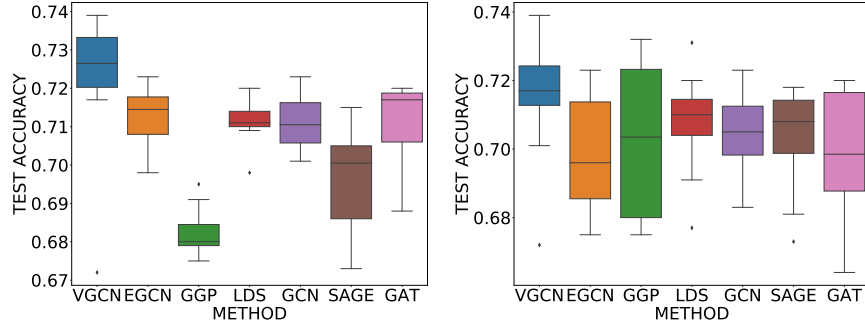

Figure 1: Test accuracy for the no-graph case on CITESEER (left) and CORA (right) across ten replications for our method (VGCN) and competing algorithms. The first three methods are Bayesian, while the others are not.

consider other Bayesian approaches to GCN, in particular, the work of [46] which herein we refer to as ensemble graph convolutional network (EGCN) and the graph Gaussian process (GGP) approach of [46]. Finally, in the adversarial case we compare against robust GCN (RGCN) of [48] that extends GCN for robustness to adversarial attacks. Overall, we believe this set of 7 benchmarks provides state-of-the-art competing algorithms that show a realistic and up-to-date evaluation of the benefits of our approach. In terms of performance metrics, throughout our experiments we use the test accuracy as given by the proportion of correctly classified test examples and use the validation accuracy for model selection on all methods. More details of the experimental set-up are given in the supplement.

## 4.2 Results in the no-graph case

Here we present the performance of our proposed method (VGCN) and the baselines' when we do not use the graph that is associated with the corresponding citations network. In this case we build a K-NNG and use it to construct a prior distribution (as described in § 2.2) for the Bayesian methods (VGCN, EGCN, GGP) or directly for the non-Bayesian algorithms (LDS, GCN, GRAPHSAGE, GAT). We see that on CITESEER (left of fig. 1) our algorithm outperforms both Bayesian and non-Bayesian methods, with the Bayesian GGP approach of [46] providing the lowest test accuracy. Although there is no clear distinction of Bayesian vs non-Bayesian methods, these results are not incredibly surprising as the non-Bayesian methods optimize the cross-entropy error directly.

The benefits of our approach are less pronounced on CORA as seen on the right of fig. 1, while we still see a marginal improvement over the other baselines. We attribute these differences between the datasets to the types of features used, BOW vs TF-IDF. Overall we can conclude that our method does manage to discover new graph parameterizations that improve performance even over methods that were specifically designed to do so, such as the LDS algorithm of [13]. We also note that, as reported in [13], a dense GCN that does not use the graph (corresponding to a multi-layer Perceptron) achieves test accuracies of 58.4% and 59.1% on CITESEER and CORA respectively.

## 4.3 Results in the adversarial setting

We consider the robustness of VGCN and the baselines in the adversarial setting where the ground truth graph has been corrupted via the removal or addition of edges. Specifically, we consider the graph poisoning setting as outlined in [4] where the graph is poisoned before model training. We limit our study on general attacks where the attacker uses an unsupervised approach to perturb network structure and is not targeted to a classification task. In [4], it was shown that poisoning the network structure reduces performance on downstream tasks and transfers across to graph convolutional methods. These graph poisoning attacks are the easiest for an attacker to deploy in practice.

We applied graph poisoning on all 3 of our datasets removing 2,000, 1,000, and 500 edges (denoted using negative values in the figures) and also adding 500, 1000, 2,000, and 5,000 edges. We generated 10 attacked graphs for each of these settings. We note that removing 5,000 edges is not actually possible on CITESEER hence we do not consider this setting. We show results for the extreme cases -2,000, 2,000, and 5,000 here and for all settings in the supplement.

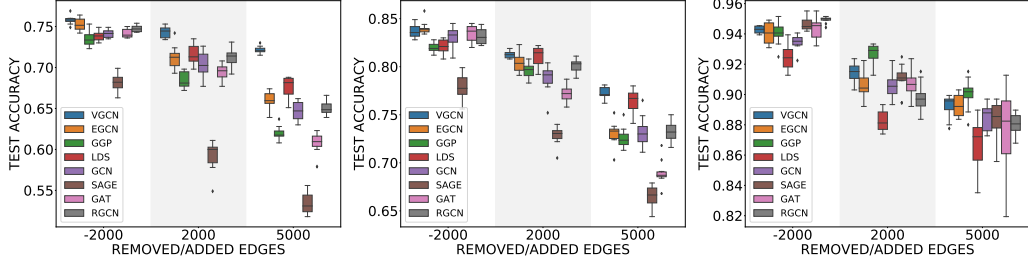

Figure 2: Test accuracy for the adversarial setting on attributed graphs CITESEER (left) and CORA (middle) and featureless graph POLBLOGS (right) when removing (negative values) or adding (positive values) edges. Our method is denoted by VGCN.

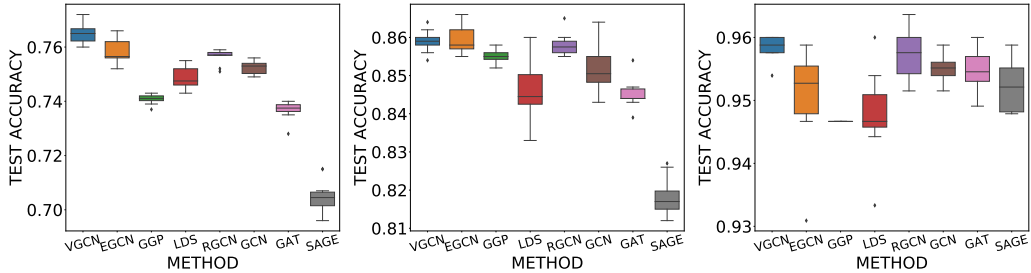

Figure 3: *From left to right*: Test accuracy for ground-truth graphs on CITESEER, CORA and POLBLOGS across ten replications for our method (VGCN) and competing algorithms.

*Citation Networks*: Figure 2 (left and middle) shows results for attacked graphs with node features, namely the two citation networks. We see that all methods perform well in the -2000 setting where the graphs are missing edges but all remaining edges are uncorrupted. The Bayesian methods VGCN and EGCN have an advantage over the others on CITESEER but the algorithms' performance is more leveled up on CORA. We note that performance for all methods degrades when false edges are added to the graphs. VGCN is the most robust especially in the extreme case of adding 5000 edges, effectively doubling the total number of edges in the graphs for both datasets. VGCN outperforms RGCN, which was specifically designed for these types of problems, especially in the cases of adding edges. Lastly, we note that variance for VGCN is lower across all graphs and datasets.

*Featureless Graphs*: Finally, we evaluate the performance of all methods on a graph without node features. Figure 2 (right) shows results for the POLBLOGS network where node features are not available. All methods perform competitively across all attacked graph settings. Surprisingly, methods such as GRAPHSAGE and graph attention network (GAT) show superior performance for the -2000 setting but have very high variance at the 5000 regime. RGCN performs best when removing edges but does poorly when adding edges due to its heavy reliance on the similarity of node features.

While the Bayesian methods, VGCN, EGCN, and GGP are the most robust on this dataset, LDS is the worst performer across all graphs. One possible explanation for this is that the lack of node features negatively affects methods that optimize the graph structure as there may not be enough information in the training data and the graph structure alone to optimize the models with higher learning capacities, i.e., more parameters. Furthermore, we note that since the ground truth graph has approximately 16,000 edges, the 5000 fake edges are only an additional 30% as compared to nearly doubling of the number of edges for the citation networks. For higher levels of noise, Bayesian methods might demonstrate higher levels of robustness.

## 4.4 Performance on ground-truth graphs and qualitative analysis

**Performance on ground-truth graphs.** Figure 3 shows the results for the ground-truth (unperturbed) graphs, where we see that VGCN can provide state-of-the-art performance, indicating that it can find better configurations even for those graphs that are believed to be most beneficial for the semi-supervised node classification task.

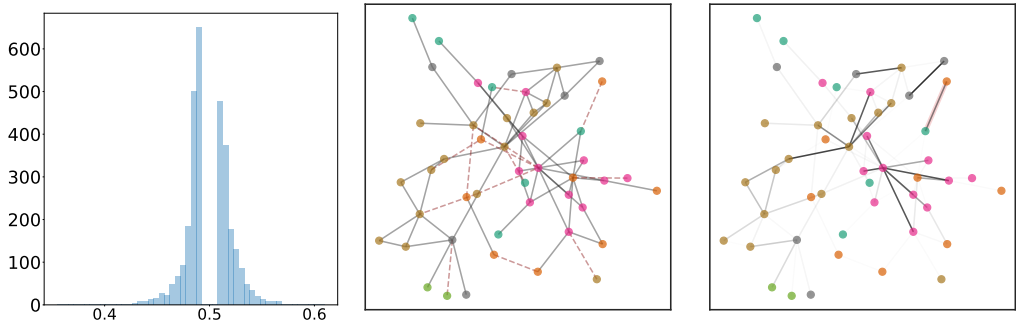

Figure 4: *Left:* Histogram of limit posterior probabilities of a link being turned on computed as the zero-temperature limit of the final variational posterior distributions over each adjacency entry (see text for details), using the CITESEER dataset under the no-graph scenario. *Middle:* A community in the original graph from the adversarial CITESEER experiment with node labels distinguished by colors and added edges denoted by red dashes. *Right:* Learned graph with edge opacity proportional to limit posterior probabilities. Added edges with probability greater than ½ are highlighted red.

**Limit posterior probabilities.** Here we analyze what our model has learned and how different the resulting posterior is with respect to the prior and its initialization during optimization. Interpreting Concrete distributions can be cumbersome so instead we use their zero-temperature property as presented in [28]. More specifically, in the zero-temperature limit, one can obtain the corresponding Bernoulli parameters of their discrete counterpart using $p(\lim_{\tau \to 0} A_{ij} = 1) = \lambda_{ij}/(\lambda_{ij} + 1)$. We refer to these probabilities as the *limit posterior probabilities*. Figure 4 (left) shows a histogram of these limit probabilities on a typical run of our model in the no-graph case for the CITESEER dataset. As the majority of the links are close to zero, fig. 4 (left) only shows those probabilities that changed significantly with respect to the prior (defined as their absolute difference being greater than 0.02). Hence, we see that in effect our algorithm manages to both decrease and increase these initial probabilities, providing evidence that it has the capacity to turn links on/off in the original graph. More precisely, the number of links with a significant change in their probabilities of being turned on was 3046. We believe this is indeed a significant amount, considering that the k-nearest neighbor graph had around 33,000 links and that the 'true' CITESEER graph has around 4600 edges.

**Learned graphs.** We illustrate the types of graphs learned by our approach by using an example from the adversarial setting experiment on CITESEER when adding edges, where our approach significantly outperformed the competing baselines. Since showing the entire graph would be unintelligible, we enumerate communities (subgraphs that are internally densely connected) that contain a good balance of node labels. In particular, we use label propagation [49] to detect the largest communities and draw a graph for each. We show an example subgraph in fig. 4 (middle and right). On the middle we denote the edges from the original graph in solid lines and the added edges in dashed red lines. On the right is the corresponding complete graph with edge opacity proportional to their limit posterior probabilities. Generally speaking, the posterior probabilities of the original edges can be expected to remain largely the same or in some cases, either amplified or attenuated to improve downstream classification accuracy. More interestingly, we see that, with few exceptions (highlighted in red), the posterior probabilities of the added edges are attenuated. We show more example subgraphs in the supplement.

## 5  Conclusion & discussion

We have presented variational graph convolutional networks (VGCNs) for semi-supervised classification, a method that generalizes the capabilities of GCNs by making them applicable in the absence of graph data and more robust to adversarial attacks. VGCN considers prior distributions over the graph along with a GCN likelihood in a joint probabilistic model and infers a graph posterior exploiting Concrete distributions. We have showcased the performance of our method on the above problems and using the ground-truth graphs. Our method can be extended beyond GCNs to use other graph neural network architectures as long as the graph is represented by an adjacency matrix and that parameter estimation in the original method can be combined easily with variational inference.

## Broader Impact

There exist numerous useful applications for graph-CNNs including e-commerce product recommendations [41], online social network recommendations [45], drug discovery [27, 32, 14], computational pharmacology [50], disease understanding [39], bioinformatics [12], finance [16], anti-money laundering [42], online hate speech classification [36], and understanding online fake news propagation [30].

Of the above applications, several can benefit from VGCN over existing graph-CNNs. Any application where there is incentive for bad actors to poison the data in order to (a) hide their activities, e.g., online hate speech, (b) control the activities of others, e.g., trick consumers into purchasing unreliable or expensive products, or (c) promote misinformation, e.g., fake news, stand to benefit from VGCN's ability to deal with adversarial attacks.

Furthermore, VGCN is directly applicable to non-graph domains via the ad hoc construction of graphs hence the benefits of graph-CNNs can be brought into domains where data is not inherently graph-structured. As we have discussed earlier, such ad hoc graph creation results in noisy graphs requiring the use of principled structure modeling for training useful predictive models.

Just like any machine learning algorithm can be used for good it can also be used for harm. Graph-CNNs and our method VGCN are not immune to such misuse. For example, understanding how VGCN deals with adversarial attacks can be used by an adversarial agent to create more robust attacks and subvert attempts at detection. Currently, we have no solution for such a general problem but we understand that this needs to be addressed in future work.

Overall, we believe that our work is beneficial to society because of the many important applications that stand to benefit from VGCN's ability to handle noise in the graph structure.

## Acknowledgments and Disclosure of Funding

We thank Harrison Nguyen for his contribution to an earlier version of this paper presented at NeurIPS 2019's Graph Representation Learning (GRL) workshop. LT is supported by an Australian Government Research Training Program (RTP) Scholarship and a CSIRO Data61 Postgraduate Scholarship. This work was conducted in partnership with the Defence Science and Technology Group, through the Next Generation Technologies Program.

## Footnotes

[2]Code available at https://github.com/ebonilla/VGCN.

[3]sample-and-aggregate (GRAPHSAGE) has been shortened to SAGE in the figures.

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
