[Supplementary Material]

# Variational Inference for Graph Convolutional Networks in the Absence of Graph Data and Adversarial Settings:
## *Supplementary Material*

**Pantelis Elinas**[*]
pantelis.elinas@data61.csiro.au
CSIRO's Data61

**Edwin V. Bonilla**[*]
edwin.bonilla@data61.csiro.au
CSIRO's Data61 and The University of Sydney

**Louis Tiao**
louis.tiao@sydney.edu.au
The University of Sydney and CSIRO's Data61

## A  Variational distributions

In this section we provide more details about the choices of variatiational distributions over the graph structure.

### A.1  Variational distribution: free vs smooth parameterizations

Similarly to the prior definition, our approximate posterior is of the form

$$q_{\boldsymbol{\phi}}(\mathbf{A}) = \prod_{ij} q_{\boldsymbol{\phi}}(A_{ij}), \text{ with } \quad q_{\boldsymbol{\phi}}(A_{ij}) = \text{Bern}(A_{ij} \,|\, \rho_{ij}), \rho_{ij} > 0, \quad\quad\quad \text{(A.1)}$$

where, henceforth, we use $\boldsymbol{\phi}$ to denote all the parameters of the variational posterior. In the case where $\rho_{ij}$ are free parameters then $\boldsymbol{\phi} = \{\rho_{ij}\}$. We refer to this approach as the *free* parameterization We have found experimentally that such a parameterization can make optimization of the evidence lower bound (ELBO) wrt $\boldsymbol{\phi}$ extremely difficult. Thus, one is forced to either use alternative representations of the posterior, or continuous relaxations of the discrete prior and posterior distributions (see appendix A.2). Intuitively, conditional independence in the posterior is a strong assumption and small changes in $\rho_{ij}$ will compete with each other to explain the data. Consequently, any continuous optimization algorithm will find it very challenging to find a good direction in this non-smooth combinatorial space. Therefore, as an alternative, it is sensible to adopt a smooth parameterization:

$$\rho_{ij} = \sigma(\mathbf{z}_i^T \tilde{\mathbf{z}}_j + b_i + b_j + s), \mathbf{z}_i, \tilde{\mathbf{z}}_j \in \mathbb{R}^{d_z}, \{b_i, s \in \mathbb{R}\}, \quad\quad\quad \text{(A.2)}$$

$i, j = 1, \ldots, N$, where $\sigma(x) \equiv (1 + \exp(-x))^{-1}$ is the logistic sigmoid function and $d_z \leq D$ is the dimensionality of the parameters $\mathbf{z}, \tilde{\mathbf{z}}$. As we see, the same representation is shared across the columns and rows of $\mathbf{A}$'s Bernoulli parameters, which addresses the combinatorial nature of the optimization landscape of the free parameterization. We note that this parameterization is referred to in the matrix-factorization and link-prediction literature as low-rank [16, 15] or dot-product [12]. In this case the variational parameters are $\boldsymbol{\phi} = \{\{\mathbf{z}_i, \tilde{\mathbf{z}}_i, b_i\}, s\}$.

### A.2  Variational distribution: discrete vs relaxed

We have defined above a variational distribution which naturally models the discrete nature of the adjacency matrix $\mathbf{A}$. Our goal is to estimate the parameters $\boldsymbol{\phi}$ of the posterior $q_{\boldsymbol{\phi}}(\mathbf{A})$ via maximization

---

[*]Joint first author.

of the ELBO. For this purpose we can use the so-called score function method [17], which provides an unbiased estimator of the gradient of an expectation of a function using Monte Carlo (MC) samples. However, it is now widely accepted that, because of its generality, the score function estimator can suffer from high variance [18].

Therefore, as an alternative to the score function estimator, we can use the so-called re-parameterization trick [11, 19], which generally exhibits lower variance. Unfortunately, the re-parameterization trick is not applicable to discrete distributions so we need to resort to continuous relaxations. In this work we use Concrete distributions as proposed by Jang et al. [8], Maddison et al. [14]. In particular, we denote our binary Concrete posterior distribution with location parameters $\lambda_{ij} > 0$ and temperature $\tau > 0$ as $q_\phi(A_{ij}) = \text{BinConcrete}(A_{ij} \,|\, \lambda_{ij}, \tau)$. Analogously, as discussed in Maddison et al. [14], in order to maintain a lower bound during variational inference we also relax our prior so that $p(A_{ij}) = \text{BinConcrete}(A_{ij} \,|\, \lambda_{ij}^o, \tau_o)$. In this case the variational parameters are the parameters of the Concrete distribution which can be, as in the discrete case, free parameters $\phi = \{\lambda_{ij}\}$ or have a smooth parameterization analogous to that in eq. A.2, i.e. $\lambda_{ij} = \exp(\mathbf{z}_i^T \mathbf{z}_j + b_i + b_j + s)$ and, consequently, $\phi = \{\{\mathbf{z}_i, b_i\}, s\}$.

## B  Binary discrete distributions

The Kullback-Leibler (KL) divergence between two Bernoulli distributions $q(a \,|\, \rho)$ and $p(a \,|\, \rho^o)$ can be computed as

$$\text{KL}\left[q(a \,|\, \rho) \,\|\, p(a \,|\, \rho^o)\right] = \rho[\log \rho - \log \rho^o] + (1 - \rho)[\log(1 - \rho) - \log(1 - \rho^o)]. \tag{B.1}$$

## C  Binary concrete distributions

In this section we give details of the re-parameterization used for the implementation of our algorithm when both the prior and the approximate posterior are relaxed via the binary Concrete distribution [14, 8].

### C.1  Summary of Bernoulli relaxation transformations

$$A_{ij} \sim \text{BinConcrete}(\lambda_{ij}, \tau) \quad \Leftrightarrow \quad A_{ij} = \sigma(B_{ij}), \quad B_{ij} \sim \text{Logistic}\left(\frac{\log \lambda_{ij}}{\tau}, \frac{1}{\tau}\right); \tag{C.1}$$

$$B_{ij} \sim \text{Logistic}\left(\frac{\log \lambda_{ij}}{\tau}, \frac{1}{\tau}\right) \quad \Leftrightarrow \quad B_{ij} = \frac{\log \lambda_{ij} + L}{\tau}, \quad L \sim \text{Logistic}(0, 1); \tag{C.2}$$

$$L \sim \text{Logistic}(0, 1) \quad \Leftrightarrow \quad L = \sigma^{-1}(U) := \log U - \log(1 - U), \quad U \sim \text{Uniform}(0, 1). \tag{C.3}$$

In summary, we have

$$A_{ij} \sim \text{BinConcrete}(\lambda_{ij}, \tau) \quad \Leftrightarrow \quad A_{ij} = \sigma\left(\frac{\log \lambda_{ij} + \sigma^{-1}(U)}{\tau}\right) \quad U \sim \text{Uniform}(0, 1). \tag{C.4}$$

### C.2  Re-parameterized ELBO

With the results above, it is easy to see that we can write the ELBO as:

$$\mathcal{L}_{\text{ELBO}}(\phi) = \mathbb{E}_{q_{\phi,\tau}(\mathbf{A})}\left[\log p_{\boldsymbol{\theta}}(\mathbf{Y}^o \,|\, \mathbf{X}, \mathbf{A}) - \log \frac{q_{\phi,\tau}(\mathbf{A})}{p_{\tau_o}(\mathbf{A})}\right] \tag{C.5}$$

$$= \mathbb{E}_{g_{\phi,\tau}(\mathbf{B})}\left[\log p_{\boldsymbol{\theta}}(\mathbf{Y}^o \,|\, \mathbf{X}, \sigma(\mathbf{B})) - \log \frac{q_{\phi,\tau}(\sigma(\mathbf{B}))}{p_{\tau_o}(\sigma(\mathbf{B}))}\right] \tag{C.6}$$

$$= \mathbb{E}_{g_{\phi,\tau}(\mathbf{B})}\left[\log p_{\boldsymbol{\theta}}(\mathbf{Y}^o \,|\, \mathbf{X}, \sigma(\mathbf{B})) - \log \frac{g_{\phi,\tau}(\mathbf{B})}{f_{\tau_o}(\mathbf{B})}\right]. \tag{C.7}$$

where

$$g_{\phi,\tau}(B_{ij}) = \text{Logistic}\left(B_{ij} \,|\, \frac{\log \lambda_{ij}}{\tau}, \frac{1}{\tau}\right), \quad f_{\tau_o}(B_{ij}) = \text{Logistic}\left(B_{ij} \,|\, \frac{\log \lambda_{ij}^o}{\tau_o}, \frac{1}{\tau_o}\right). \tag{C.8}$$

Table 1: Datasets used in the experiments. Train/Valid/Test correspond the the training/validation/test set sizes. Label rate refers to the ratio of the training set size over the total number of nodes.

| Dataset | Type | Nodes | Edges | Classes | Features | Train/Valid/Test | Label rate |
|---------|------|-------|-------|---------|----------|------------------|------------|
| CORA | Citation network | 2,708 | 5,278 | 7 | 1,433 | 140/500/1,000 | 0.052 |
| CITESEER | Citation network | 3,327 | 4,676 | 6 | 3,703 | 120/500/1,000 | 0.036 |
| POLBLOGS | Blog network | 1,222 | 16,717 | 2 | N/A | 122/275/825 | 0.10 |
| PUBMED | Citation network | 19,717 | 44,338 | 3 | 500 | 60/500/1,000 | 0.003 |

## C.3 Importance-weighted ELBO

For the relaxed version of our algorithm (that uses binary Concrete distributions), in which we cannot compute the KL term in the ELBO analytically, we use the importance-weighted ELBO, which has been shown to perform better than the standard ELBO, be a tighter bound of the marginal likelihood and related to variational inference in an augmented space [2, 5]:

$$\text{IW-ELBO} = \sum_{\mathbf{y}_n \in \mathbf{Y}^o} \text{LME}_{\mathbf{A}_{1:S}} \left[ \log p(\mathbf{y}_n | \mathbf{X}, \mathbf{A}) - \frac{1}{|\mathbf{Y}^o|} \log \frac{q_{\phi,\tau}(\mathbf{A})}{p_{\tau_o}(\mathbf{A})} \right], \qquad \text{(C.9)}$$

where $\text{LME}_{\mathbf{A}_{1:S}}(h(\mathbf{A}))$ is the log-mean-exp operator of function $h(\mathbf{A})$ over samples of $\mathbf{A}$, i.e. $\text{LME}_{\mathbf{A}_{1:S}} = \log \frac{1}{S} \sum_{s=1}^{S} \exp(h(\mathbf{A}^{(s)}))$ with $\mathbf{A}_{1:S} \equiv (\mathbf{A}^{(1)}, \ldots, \mathbf{A}^{(S)})$ and $\mathbf{A}^{(s)} \sim q_{\phi,\tau}(\mathbf{A})$.

# D  Implementation and computational complexity

We implement our approach using TensorFlow [1] for efficient GPU-based computation and also use some components of TensorFlow Probability [4][2]. The time complexity of our algorithm can be derived from considering the two main components of the ELBO in eq. 5, namely the KL divergence term and the expected log likelihood (ELL) term. We focus here on one-hidden layer graph convolutional network (GCN) (apart from the output layer) with dimensionality $Q \equiv Q^{(1)}$ along with a smooth (dot product) parameterization of the posterior. We recall that $N$ is the number of nodes, $D$ is the dimensionality of the input features $\mathbf{X}$, $d_z$ is the dimensionality of posterior parameters $\mathbf{Z}$, $C$ is the number of classes and $S$ is the number of samples from the variational posterior used to estimate the required expectations.

**KL divergence term**: We require to compute $\mathcal{O}(N^2)$ individual KL divergences, which can be trivially parallelized. In the case of the smooth parameterization, for both the discrete and the relaxed cases, we need to compute the dot-product between the latent representations for each each $A_{ij}$ which is $\mathcal{O}(d_z)$ and gradient information must be aggregated for each $\mathbf{z}_i, \tilde{\mathbf{z}}_i$. While for a discrete posterior these individual KL terms can be computed exactly (as shown in appendix B), for the continuous relaxation we need to resort to MC estimation over $S$ samples. Aggregation over samples can also be parallelized straightforwardly.

**ELL term**: Computing the ELL using a 2-layer GCN as in eq. 1 requires $\mathcal{O}(NDQ + S(NQC + N^2Q + N^2C))$ for the continuous case. However, in the discrete case it only requires doing a forward pass over the standard GCN architecture $S$ times, hence being linear in the number of edges, i.e. $\mathcal{O}(S|\mathcal{E}|DQC)$, where $|\mathcal{E}|$ is the expected number of edges sampled from the posterior, assuming sparse-dense matrix multiplication is exploited.

# E  Datasets

Table 1 gives details of the datasets used in our experiments. Training/Valid/Test refer to the default training/validation/test set sizes. However, as mentioned in § 4, we adopt a similar approach to that of Franceschi et al. [6] where the training set is augmented with approximately 50% of the validation set.

# F   Full details of experimental set-up

Unless stated explicitly below, all the optimization-based methods were trained up to a maximum of 5,000 epochs using the ADAM optimizer [10] with an initial learning rate of 0.001. Hyper-parameter exploration was done via grid search and model selection carried out via cross-validation using the accuracy on the validation set. For standard GCN and our method we use a two-layer GCN as given in eq. 1 with a 16-unit hidden layer. We train standard GCN as done by Kipf and Welling [13] so as to minimize the cross-entropy loss, using dropout and L2 regularization, Glorot weight initialization [7] and row-normalization of input-feature vectors. As with our method, we set the dropout rate to 0.5 and the regularization parameter in $\{5 \times 10^{-3}, 5 \times 10^{-4}\}$.

For our method, we carried posterior estimation over the adjacency matrix and MAP estimation of the GCN-likelihood parameters so as to maximize the ELBO in eq. 5. Hyper-parameters for GCN-estimation were the same as above. To construct the prior over the adjacency matrix we followed the procedure explained in § 2.2 with $\bar{\rho}_1 = \{0.25, 0.5, 0.75, 0.99\}$, $\bar{\rho}_0 = 10^{-5}$, $\tau_o = \{0.1, 0.5\}$, $\tau = \{0.1, 0.5, 0.66\}$ and $\beta = \{10^{-4}, 10^{-3}, 10^{-2}, 1\}$. We initialized the posterior to the same smoothed probabilities in the prior and used $S_{\text{train}} = 3$ and $S_{\text{pred}} = 16$ samples for estimating the required expectations for training and predictions, respectively. In the no-graph case all the methods explored k-nearest neighbor graphs (K-NNGs) with $k = \{10, 20\}$ and distance metrics $\{\text{cosine}, \text{Minkowski}\}$.

For learning discrete structures (LDS) we used the code provided by the authors[3], which carries out bilevel optimization of the regularized cross-entropy loss and does model selection based on the validation accuracy using grid search across a range of parameters such as learning rates (for inner and outer objectives), number of neighbors and distance metrics. Similarly, for graph Gaussian process (GGP) and ensemble graph convolutional network (EGCN) we used the code provided by the authors[4].

For robust GCN (RGCN) we used the code provided by the authors[5]. For all experiments, we used the default parameters as described in [22]. Specifically, we used 2 hidden graph convolutional layers with 32 units each and dropout 0.6. All models were trained for a maximum 200 epochs with early stopping (20 epochs patience) using the ADAM optimizer and an initial learning rate of 0.01.

We used our own implementation of graph attention network (GAT) and sample-and-aggregate (GRAPHSAGE). GAT used an architecture identical to the one described in Veličković et al. [21]. The first layer consists of $K = 8$ attention heads computing $F = 8$ features, followed by an exponential linear unit (ELU) nonlinearity. The second layer is used for classification that computes $C$ features (where $C$ is the number of classes), followed by a softmax activation. $L2$ regularization with $\lambda = 0.0005$ and 0.6 dropout was used. The implementation of GRAPHSAGE used mean aggregator functions and sampled the neighborhood at a depth of $K = 2$ with neighborhood sample size of $S_1 = 25, S_2 = 10$ and batch size of 50. The model was trained with 0.5 dropout and $L2$ regularization with $\lambda = 0.0005$.

# G   Complete set of results for the adversarial setting

Here we include the complete set of experiments for the 7 attacked graphs removing 2000, 1000, and 500 edges as well as adding 500, 1000, 2000, and 5000 edges to the ground truth graphs. Figure 1 shows the results for all graphs.

On the citation networks, CITESEER and CORA, our proposed variational graph convolutional network (VGCN) outperforms all other Bayesian and non-Bayesian methods, especially in the case of adding edges. On the POLBLOGS network that is lacking node features, all methods perform similarly with GGP having a small edge in the cases of adding 2000 and 5000 edges. Our method, displays the lowest variance across all datasets.

Figure 1: Results for the adversarial setting on attributed graphs CITESEER (top), CORA (middle), and (featureless graph) POLBLOGS (bottom): Accuracy on the test set across ten attacked graphs at each attack setting such that negative values indicate removing edges and positive values adding edges. We compare our method (VGCN) with competing algorithms.

## H  Low-rank vs free parameterizations

Figure 2 compares the low-rank parameterization vs the free parameterization of our model, where we used a latent representation of dimensionality $d_z = 100$. Our goal is to analyze whether a much more compact representation can yield similar results to those obtained by the free parameterization. For the citation networks used, the number of latent variables with the low-rank parameterization is $N \times 2 \times d_z \approx 6 \times 10^5$, whereas with the free parameterization we have $N \times (N-1)/2 \approx 4.5 \times 10^6$ latent variables. i.e. in this setting, the low-rank parameterization has an order of magnitude fewer latent variables. We see, in fig. 2, that although the low-rank parameterization can in some cases

achieve a performance close to that of the free-parameterization, it also has a much higher variance and in most cases the resulting solution is considerably poorer. Nevertheless, we believe that factors such as initialization can improve the performance of the low-rank parameterization significantly and leave a much more thorough study of this for future work.

## I    Discrete vs relaxed

Besides the number of latent variables used to represent the posterior, we also want to investigate the effect of using the discrete Bernoulli distributions along with the score function estimators versus the relaxed binary Concrete distributions and the reparameterization trick. Figure 3 shows the performance of these two approaches on the citations networks under study and the no-graph case when using only $S = 3$ posterior samples for prediction. We see that there is not much difference between the two approaches, although the relaxed version exhibits some outliers on CITESEER (which is ameliorated when using $S = 16$ samples) and the discrete version has slightly higher variance on CORA. However, as we see in fig. 4, the relaxed version converges much faster than the discrete version, hence our selection of the former for our main results.

Figure 5 shows results of the discrete vs relaxed parameterization in the adversarial setting for the CITESEER and CORA datasets. Both models were trained for a maximum 5000 epochs with hyperparameter optimization and model selection as described in appendix F. In the adversarial setting, there is an advantage to using the relaxed parameterization as it clearly outperforms the discrete one across all attack settings and both datasets. The difference in performance is more pronounced for CITESEER. Lastly, the relaxed parameterization exhibits lower variance across all graphs.

## J    The Effect of the number of posterior samples on predictions

Figure 6 shows results for our model when using $S = 3$ and $S = 16$ samples from the posterior when making predictions. We observe that, as expected, using more posterior samples does improve performance and that the additional gains of using more samples are worthwhile if the computational constrains can be satisfied.

## K    The influence of the KL term during training

As mentioned in the main paper, we scaled the KL term by a dampening factor $\beta < 1$ so that it does not dominate the likelihood term in the ELBO. We have analyzed this KL-dampening factor on the adversarial experiments across all datasets. Figure 7 shows how frequently each factor was selected through cross-validation with $\beta = \{1, 10^{-2}, 10^{-3}, 10^{-4}\}$ being selected $\{32\%, 35\%, 21\%, 12\%\}$ of the time, respectively. Along with the performance benefits shown in the main paper, this confirms that KL regularization resulting from variational inference does have an effect. Lastly, we have found that even when $\beta$ is very small, the KL term still has an effect as it can be several orders of magnitude larger than the ELL.

## L    Posterior analysis in the adversarial setting

Similar to the analysis in § 4.4 of the main paper for the no-graph case, here we look at the posterior changes for a representative experiment in the adversarial setting. Figure 8 shows the difference between the final posterior probabilities obtained by our algorithm and the prior probabilities, which were also used to initialize the posterior. We see that our model manages to effectively turn off/turn on a significant number of links.

## M    Additional examples of learned graphs

Following on from § 4.4 of the main paper, we visualize additional communities of the CITESEER citation network with added edges and the corresponding latent graph inferred using our approach. We show four communities in fig. 9. As before, on the left we denote the edges from the original graph in solid lines and the added edges in dashed red lines; on the right is the corresponding complete graph with edge opacity drawn proportionally to the limit posterior probabilities.

Figure 2: Results in the no-graph case for CITESEER (left) and CORA (right)—low-rank vs free parameterization: The test accuracy of our method (VGCN) using a low-rank parameterization and a free parameterization, with the latent dimensionality of the low-rank parameterization $d_z = 100$. As with the results in the main paper, In both cases we used $S_{\mathrm{pred}} = 16$ posterior samples for predictions.

Figure 3: Results in the no-graph case for CITESEER (left) and CORA (right)—discrete vs relaxed: The test accuracy of our method (VGCN) using a free parameterization with discrete Bernoulli distributions and relaxed binary Concrete distributions. Unlike the results in the main paper, in both cases we used $S_{\mathrm{pred}} = 3$ posterior samples for predictions.

Figure 4: Results in the no-graph case—convergence of discrete vs relaxed approaches on CITESEER: Convergence of the free parameterization with the discrete and the relaxed version using $S_{\mathrm{pred}} = 3$ posterior samples for predictions.

Figure 5: Results in the adversarial setting for CITESEER (top) and CORA (bottom) —discrete vs relaxed: The test accuracy of our method (VGCN) using a free parameterization with discrete Bernoulli distributions (VGCN-discrete) and relaxed binary Concrete distributions (VGCN-relaxed). In both cases we used $S_{\text{pred}} = 3$ posterior samples for predictions.

Figure 6: Results in the no-graph case for CITESEER (left) and CORA (right)—effect of the number of samples: The test accuracy of our method (VGCN) (with freely parameterized relaxed binary Concrete distributions) using $S \equiv S_{\text{pred}} = 3$ and $S \equiv S_{\text{pred}} = 16$ posterior samples for predictions.

Figure 7: Histogram of KL-dampening factor $\beta$ selected using cross-validation for experiments using the CITESEER, CORA, and POLBLOGS datasets, where $\beta = 1$ accounts for no dampening of the KL term in the ELBO.

Figure 8: Posterior changes in the adversarial setting on CITESEER. The difference in the limit probabilities computed as the zero-temperature limit of the final variational posterior distributions over each adjacency entry. Only showing those probabilities that changed significantly from the prior, which had a maximum value of $0.25$ and a minimum value of $10^{-5}$. The total number of changed probabilities was $9,504$.

Furthermore, let $E'$ denote the set of added edges. If we have inferred a posterior that suppresses the negative influence of these added edges, we would expect that $\{i, j\} \in E'$ implies $p(A_{ij}) < 1/2$. On the right, we highlight in red every edge $\{i, j\} \in E'$ where $p(A_{ij}) \geq 1/2$. We can see that such cases are few and far between, even in communities predominantly consisting of added edges (e.g. row 3).

## N   Results using random data splits

It was argued in [20] that model evaluation using pre-existing data train/validation/test splits produces overconfident estimates of a GNN model's performance. It is thus suggested that random splits of the data should be instead used. Here we have repeated the experiments outlined in section G using random splits and show the results in fig. 10. We have used the same random splits to evaluate all competing methods.

Comparing the results shown in fig. 1 and fig. 10, we notice that there is a small drop in performance for all methods and across all attack settings. However, overall our VGCN method continues to outperform the others especially in the setting of adding a large number of false edges. The EGCN method has a small advantage over VGCN when removing 2000 and 1000 edges on both datasets; however, we note that the variance of EGCN has also increased considerably when compared to the results using the fixed splits as shown in fig. 1. Overall, our original conclusions about the benefits of VGCN in the case of attacked graphs remain true regardless of how the given data is split for training and validation.

## O   Mini-batch training

We explored mini-batch training employing the approach of [3]. Mini-batch training permits applying our method to larger datasets where the full-batch method would result in out-of-memory errors. An additional benefit is also an order reduction in the number of model parameters through a block-diagonal approximation of the given graph; this approach is referred to as vanilla cluster-GCN in [3]. We decompose the auxiliary graph into $m$ non-overlapping subgraphs using the METIS [9] algorithm. Then, we optimise the VGCN parameters using mini-batch SGD considering each subgraph as a mini-batch.

Figure 11 shows the performance of our method using mini-batch training with a block-diagonal approximation on the PUBMED dataset in the adversarial setting. The statistics for the PUBMED dataset are shown in Table 1. We can see that this is a much larger dataset than CORA and CITESEER. In the adversarial setting, we consider attacks that add or remove edges proportionally to the total number of edges in the ground truth graph such that we remove approximately 50% of the edges or add 50% or 100% of edges. For these experiments we limited the search space for the hyper-parameters to the following: GCN regularisation in $\{5 \times 10^{-3}, 5 \times 10^{-4}\}$, $\bar{\rho}_1 = \{0.25, 0.5, 0.75\}$, $\beta = \{10^{-2}, 10^{-3}\}$, $\tau_o = \{0.5\}$, and $\tau = \{0.5, 0.66\}$. Finally, we reduced the given graph to 20 non-overlapping graphs.

In fig. 11 we compare VGCN with GCN, GAT, and RGCN. We see that in the case of attacks that add edges to the graph, our VGCN method outperforms all others. In the case of attacks that remove edges, all methods have similar performance although VGCN displays a small performance drop and higher variance. This can easily be explained as an artifact of the block-diagonal approximation that forces VGCN to consider a much smaller set of edges than the other methods. That is, the number of edges removed are both the attacked ones as well as the between-subgraph edges.

## Footnotes

[2] Code available at https://github.com/ebonilla/VGCN.

[3]https://github.com/lucfra/LDS-GNN.

[4]https://github.com/yincheng/GGP for GGP and https://github.com/huawei-noah/BGCN for EGCN.

[5]https://zw-zhang.github.io/files/2019_KDD_RGCN.zip

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

Figure 9: *Left:* Communities in the original graph from the adversarial CITESEER experiment with node labels distinguished by colors and added edges denoted by red dashes. *Right:* Learned graph with edge opacity proportional to limit posterior probabilities. Added edges with probability greater than ½ are highlighted red.

Figure 10: Results for the adversarial setting on attributed graphs CITESEER (top), CORA (bottom) using random splits of the data into train/validation/test sets: Accuracy on the test set across ten attacked graphs at each attack setting such that negative values indicate removing edges and positive values adding edges. We compare our method (VGCN) with competing algorithms.

Figure 11: Results using mini-batch training with a block-diagonal approximation of the auxiliary graph on the Pubmed dataset in the adversarial setting.