[Reviews · NeurIPS 2020]

Review 1

Summary and Contributions: This paper improves GCN in the context of absence/corruption of an input graph. To achieve this, the author proposed a probabilistic framework with a prior over the graph structure. The inference is done with a mean-field variational inference, coupled with Concrete relaxation. Empirical results validate the usefulness of the proposed methods.

Strengths: The paper is mathematically correct, and the empirical evaluation is sound. It is relevant to the NeurIPS community, especially for the audience interested in improving graph structure for graph neural networks.

Weaknesses: Novelty is not significant. I believe it is a standard practice of applying probabilistic framework and variational inference into GCN. See more comments in additional feedback.

Correctness: Yes, claims, method, and empirical methodology are correct.

Clarity: Yes, the paper is well written.

Relation to Prior Work: Yes.

Reproducibility: Yes

Additional Feedback: (1) Too many free parameters to store and optimize. Though the author argued freely-parameterized variational posterior is better than a low-rank parameterization. It will contain O(N^2) free parameters. Consequently, I don’t think it scales well with the graph size. (2) The scaling parameter \beta for the KL regularization term in ELBO could be too small. The value is cross-validated from \beta = {10^−4, 10^−3, 10^−2, 1} (line 98, supplementary). These numbers could be too small (except 1), leading the KL regularizer to have minor influence. (3) For the adversarial setting, results from figure 3 do not provide evidence that VGCN is better than other competing methods. (4) Figure 2 is not a clear way to compare different methods under three experimental settings. At least, the size of the picture could be larger. (5) Legend for Figure 1 is missing. >>> Post rebuttal: The authors’ feedback has largely addressed my concerns, regarding the novelty and experiments. I would like to update my score for an accept.


Review 2

Summary and Contributions: The authors propose a method for jointly estimating the the graph posterior and the parameters of a graph convolutional network. The motivation is a framework that is able to address the situation where an input graph is noisy/unreliable and a target of adversarial perturbations. The paper introduces a novel joint probabilistic model that places a prior over the graph structure along with a GCN-based conditional likelihood.

Strengths: Overall, this is a solid contribution to the growing literature of jointly learning the graph structure (or parameterizations of the graph structure) and the parameters of a GCN/GNN. The method is novel and the results look impressive. The empirical results are all in all very good and the evaluation was fair (with some exceptions, see below).

Weaknesses: I have a minor quibble with the way the introduction is written. The reader might think that the authors are the first to propose a method for cases where the graph structure is incomplete/missing and where said structure (in some form) is learned jointly with the parameters of a GCN/GNN. While prior work is fairly discussed in the related work section, I would expect the intro to read more as setting the stage for work that addresses a problem that has already been addressed by a number of papers. Instead the authors mention only GCNs, the Conrete distribution, and adversarial graph perturbations as related work in the introduction. The experiments (especially the setup) are impressive and show a clear advantage of the proposed method. I'm not sure the experiments were entirely fair to some of the more related baselines whose code was run without adjustments. For instance, moving to a graph without features and simply running other baselines without additional tuning is not 100% fair as most prior work has been evaluated with graphs with features. Here I see a bit of an unfair advantage of the proposed method as (also according to the paper) more tuning took place. One experiment I would have liked to see is a kNN graph with less than 10 neighbours (so a very sparse graph) for the no-graph experiments. But since you base this on prior work, I think it is justified not to have it. It would just be interesting. Overall though, the experimental results are solid to demonstrate that the proposed method is useful and has impressive performance across a range of datasets.

Correctness: Yes

Clarity: Yes

Relation to Prior Work: Is fairly discussed, all in all. Why do you think that LDS cannot incorporate prior knowledge? Or, in other words, what do you mean by prior knowledge? LDS parameterizes the edges with Bernoullis (exactly as your work) and these are initialized with a given graph structure.

Reproducibility: Yes

Additional Feedback:


Review 3

Summary and Contributions: This paper introduces Variational Graph Convolutional Networks (VGCNs), a framework for jointly learning a posterior over the graph structure, while performing message passing operations on the graph predicted by the inference model (posterior). The framework assumes that each edge is parameterized separately, hence this approach is tested only on transductive, semi-supervised node classification tasks. In all demonstrated experiments, the approach is more robust than well-chosen baselines and works well even in the absence of externally provided graph structure.

Strengths: Jointly learning the graph structure and the neural message passing model is a promising approach to overcome current limitations of graph neural networks, and hence this paper is highly relevant to the NeurIPS community. The proposed approach is novel. The claims are sound and the empirical evaluation is carefully executed. Overall, this is a very solid paper.

Weaknesses: There is not much to be criticised about this paper. It is well-written and well-executed. In terms of novelty, the approach is relatively straightforward (which is not a bad thing!), especially given some earlier related work that utilizes a very similar variational inference approach over discrete edges in a graph (in the context of GNNs), which the authors might have overlooked: Neural Relational Inference for Interacting Systems (Kipf et al., ICML 2018). I would further recommend to highlight a core limitation of the approach: it only works in transductive settings where all entities are known at training time. It would be good if the authors could comment on alternative modeling choices that would allow the model to be applied in inductive settings. The paper would also be stronger if it had some qualitative analysis of the learned graphs and a more diverse set of benchmark datasets / application domains. UPDATE: I would like to thank the authors for their response. I still think that this paper should be accepted.

Correctness: The claims, method, and the empirical methodology appear to be correct.

Clarity: The paper is generally well-structured and well-written.

Relation to Prior Work: Related work is adequately discussed, apart from the line of work mentioned above.

Reproducibility: Yes

Additional Feedback: Q1: What do the learned graphs look like? Is the process useful for interpretability? Q2: Currently, every edge is explicitly parameterized in the posterior, there is no parameter sharing. This limits applicability to transductive settings where all the entities are known at training time. Could this framework be extended to inductive settings where the edge posterior is a function of e.g. node embeddings? This is a setting more in line with the variational edge inference model in Neural Relational Inference (Kipf et al., ICML 2018), and it would be interesting to discuss the differences to this approach.


Review 4

Summary and Contributions: This paper introduces a model for semisupervised classification in which a set of entities are associated to binary labels (most of which are unobserved), and the entities can have observed features, or an observed network encoding the relationship between those entities (though either one or the other can be missing). In particular, this paper focus in improving the prediction performance when the network is either unobserved or corrupted. For that goal, a joint probabilistic model for the network and the response is presented, which is based on a graph convolutional network to relate the features and network with the response, and a prior distribution for the network. The model infers the network structure via a variational inference algorithm, and use the posterior distribution to infer the unknown responses. Finally, the model is evaluated on real work data, suggesting that this approach is effective in prediction compared to other methods, most of which cannot deal with network corruption.

Strengths: Overall, I think this is a good contribution that addresses a novel problem, since most of the previous work assumes that the network is noiseless, which is usually unrealistic. The model introduced by the authors could be also of interest for performing inferences about the network itself with a distribution informed by the features and observed responses. The advantage of this methodology is clearly reflected on the experiments, which show that the new method significantly outperforms other competitors, especially when the number of corrupted edges is large (Fig. 2)

Weaknesses: UPDATE: Thanks to the authors for the response to my comments. My concerns about scalability of the method have been addressed, and I still think that the paper should be accepted. ========================================================= The authors provide some discussion on the computational complexity, but there is no empirical evaluation of that. The authors mention that the algorithm requires to evaluate O(N^2) individual KL divergences (which are presumably not sparse), and although it can be parallelized, this number can be costly, especially in terms of memory, when N is large.

Correctness: The derivation of the algorithm and the techniques used in the paper seem correct and appropriate. The empirical methodology also appears to be correct.

Clarity: I found this paper very clearly written and easy to read. All the model and algorithmic decisions are motivated to solve specific challenges, so even if the reader is unfamiliar with the literature it is easy to understand the derivation of the methodology.

Relation to Prior Work: The authors provide a clear discussion of other semi-supervised algorithms for graphs, as well as comparison with state-of-the-art methods in the experiments.

Reproducibility: Yes

Additional Feedback: The paper discuss in the abstract that the method can work in the absence of graph data. While this is practically true, it seems that the method does need to have an informative graph since when this is absent, it creates a graph from the observed features using knn. That makes me wonder whether the method will still perform well when there is actually no graph data (for example, if the method is feed with a flat network) or the graph is non-informative (a network that is completely random). Although this work is mainly focused on semi-supervised prediction, I wonder if there is a way to use the generative model for the network to produce inferences about the network itself. For example, it can be interesting to identify corrupted edges, or link prediction.

[Author Response · NeurIPS 2020]

We thank the reviewers for the helpful feedback. With the exception of Reviewer #1, the reviewers agree that our approach is novel and presents a solid contribution. In particular, Reviewer #2 says that "this is a solid contribution" and that "the method is novel and the results look impressive". Reviewer #3 comments that "the proposed approach is novel" and that "this is a very solid paper". Finally, Reviewer #4 highlights that "this is a good contribution that addresses a novel problem" and that "the advantage of the methodology is clearly reflected in the experiments".

**Reviewer #1**: *(1) Novelty is not significant*. We respectfully but strongly disagree. Our contribution is significantly different to standard practice as we consider a prior distribution over the *graph* structure and use it to address specific challenges. This has been largely overlooked in the previous literature. We respectfully refer the reviewer to our summary of the other reviewers' feedback above and their corresponding detailed comments, where they agree that our approach is novel and presents a solid contribution.

*(2) Too many free parameters*. We also present a compact low-rank parameterization (LRP) of the posterior. We discuss this in the supplement, §H.1. In some cases, the LRP achieves similar performance but exhibits much higher variance, hence requiring more epochs to converge. Furthermore, as mentioned in §5, our method can be combined with other graph neural network approaches, especially with those designed to improve the efficiency of GCNs. To illustrate this point, we have recently begun development of a more scalable extension based on Cluster-GCN (Ciang et al); early experiments on PUBMED indicate that our approach, with an order-of-magnitude less parameters, can perform similarly or better than scalable competing benchmarks, while other methods such as LDS threw out-of-memory errors.

*(3) Scaling for KL too small and having a minor influence*. We have analyzed the KL-dampening factor selected through cross-validation on the adversarial experiments across all datasets. It turns out that $\beta = \{1, 0.01, 0.001\}$ get selected $\{32\%, 35\%, 21\%\}$ of the time, while $\beta = 10^{-4}$ only $12\%$. In addition, the performance benefits in the main paper show that the KL regularization resulting from variational inference does have an effect. Lastly, we have found that even when $\beta$ is very small, the KL term still has an effect as it can be several orders of magnitude larger than the ELL.

*(4) Method not the best in Fig. 3*. We believe ML papers should also showcase when a proposed approach does not work best and this was our intention in Fig. 3. This provides some guidance on when other methods might be preferable ("no free lunch"). Nevertheless, in this experiment our method is still very competitive and does not suffer from catastrophic performance degradation when node features are not available.

**Reviewer #2**: *(1) Prior work in intro*. We thank the reviewer for this suggestion and we will make the first part of the intro more consistent with the subsequent related work in Sec 1.1.

*(2) Tuning of baselines*. We would like to clarify that we tuned the baselines across all experiments including the featureless experiment. This was done to the degree to which the reference implementations allowed us to tune them and followed the recommendations of the original papers. For example, as described in the supplement, we tuned GCN similarly to our method and the LDS implementation adopts a very thorough cross-validation procedure. For other baselines such as graphSage we got around the high-variance estimate problem by using multiple predictions, which provided better performance.

*(3) Prior knowledge in LDS*. Although LDS includes a generative model, the initial probabilities use a deterministic distribution (their Algorithm 1) and do not play an explicit role in the objective function being optimized, i.e. there is no prior constraining the search-space over graphs. Using different initial probabilities will not achieve a similar effect to that obtained in our probabilistic framework. We will expand more on this in the final version.

**Reviewer #3**: *(1) Alternative choices for inductive setting*. This is indeed an exciting avenue of research that we are pursuing based on non-linear matrix factorization approaches using Gaussian processes (GPs). Incorporating GP priors over graphs can enable one to generalize to unseen nodes.

*(2) Qualitative analysis*. To qualitatively analyze our approach, we examine densely-connected subgraphs of the CITESEER graph used in the experiment shown in Fig. 7 of the supplement (i.e. adding edges). Preliminary results in the given figure (node colors indicate node labels; left: original corrupted graph with red dashed lines indicating added edges; right: learned graph shown with edge opacities proportional to posterior probabilities; best seen zoomed in on screen) indicate that, with a few exceptions, the posterior probabilities of the  *added* edges are indeed attenuated. We will expand on this analysis in the final version.

*(3) Neural relational inference (NRI)*. Thank you for the reference. NRI (Kipf et al, ICML 2018) addresses the problem of learning the interactions between components in a dynamical system given their trajectories, i.e., the entities evolve over time. Besides the similarities wrt using variational autoencoders during inference to learn these interactions, their focus is very different to ours but we are happy to cite it and expand on this in the final version.

**Reviewer #4**: *(1) Computational complexity*. Please see Reviewer #1, item (2) above.

*(2) Using the model for inferences on the network itself*. This is an interesting point. The posterior probabilities can, indeed, be used to make inferences about the network. However, one must be cautious when comparing these to the ground truth graph, as the loss being optimized focuses on the classification problem and does not explicitly include a suitable link-prediction loss. Thus, the given links enter the objective only as a soft regularizer through the KL term. Nevertheless, we refer the reviewer to our comment on qualitative analysis above, Reviewer #3, item (2).

[Meta-Review · NeurIPS 2020]

The reviews agree unanimously that this is an interesting paper and should be published, and I concur. Concerns and questions raised in the reviews have been properly addressed in the author response. Please make sure to add the additional work promised in the rebuttal (prior knowledge in LDS, qualitative analysis) before submitting for camera.